# Outcome of T2 Glottic Cancer Treated with Radiotherapy Alone or Concurrent Chemo-Radiotherapy

**DOI:** 10.3390/cancers17040712

**Published:** 2025-02-19

**Authors:** Po-Ying Peng, Chien-Yu Lin, Chun-Ta Liao, Hung-Ming Wang, Shiang-Fu Huang, Yao-Te Tsai, Chang-Hsien Lu, Fu-Min Fang, Meng-Hung Lin, Miao-Fen Chen, Wen-Cheng Chen

**Affiliations:** 1Department of Radiation Oncology, Chang Gung Memorial Hospital at Linkou, Chang Gung University, No. 5, Fu-Hsing ST., Kwei-Shan, Taoyuan 333, Taiwan; d29949@cgmh.org.tw (P.-Y.P.); qqvirus@cgmh.org.tw (C.-Y.L.); 2Department of Otorhinolaryngology, Head and Neck Surgery, Chang Gung Memorial Hospital at Linkou, Chang Gung University, Taoyuan 333, Taiwan; liaoct@cgmh.org.tw (C.-T.L.); bigmac@adm.cgmh.org.tw (S.-F.H.); 3Department of Medical Oncology, Chang Gung Memorial Hospital at Linkou, Chang Gung University, Taoyuan 333, Taiwan; whm526@cgmh.org.tw; 4Department of Otorhinolaryngology, Head and Neck Surgery, Chang Gung Memorial Hospital, Chiayi, Chang Gung University, Chiayi 613, Taiwan; s881057@cgmh.org.tw; 5Department of Medical Oncology, Chang Gung Memorial Hospital, Chiayi, Chang Gung University, Chiayi 613, Taiwan; q12014@cgmh.org.tw; 6Department of Radiation Oncology and Proton and Radiation Therapy Center, Kaohsiung Chang Gung Memorial Hospital, Chang Gung University College of Medicine, Kaohsiung 833, Taiwan; fang2569@cgmh.org.tw; 7Health Information and Epidemiology Laboratory, Chang Gung Memorial Hospital, Chiayi 613, Taiwan; mattlin@cgmh.org.tw; 8Department of Radiation Oncology, Chang Gung Memorial Hospital, Chiayi, Chang Gung University, No. 6, Chia-Pu Rd., Chiayi 613, Taiwan

**Keywords:** glottic carcinoma, T2, chemo-radiotherapy, radiotherapy, outcomes

## Abstract

Current evidence suggests that radiotherapy (RT) provides excellent locoregional control and survival rates, exceeding 90% at five years for stage I glottic cancer, particularly when hypo-fractionated regimens are employed. However, outcomes for stage II disease are less favorable, even with modifications to fractionation. In this retrospective study, we analyzed oncological outcomes in a cohort of 121 patients with T2N0M0 glottic cancer in an effort to identify the most effective treatment modality. Our findings indicate that concurrent chemoradiotherapy (CCRT) is the most effective treatment for this patient population. Specifically, the 5-year and 10-year locoregional control rates for patients receiving CCRT were 88.5% and 83.2%, respectively, compared to 72.8% and 69.6% for those treated with definitive RT alone (adjusted hazard ratio: 0.30, 95% confidence interval 0.12–0.76, *p* = 0.011). Notably, the most significant treatment effects were observed in patients with subglottic extension and vocal cord mobility impairment, which are established poor prognostic indicators for glottic cancer. Conclusion: CCRT improves local control, recurrence-free survival, and overall survival in T2N0M0 glottic cancer, albeit with high toxicity.

## 1. Introduction

Laryngeal cancer accounts for 10% of head and neck cancers, with glottic cancers comprising two-thirds of cases in our country [1]. Over 90% are squamous cell carcinomas. Tumors involving the vocal cords often cause hoarseness, enabling early diagnosis in localized stages. Early-stage glottic cancers (T1 and T2) rarely spread to cervical lymph nodes due to limited lymphatic drainage, allowing effective treatment with definitive radiotherapy (RT) or transoral laser microsurgery (TLM), both achieving excellent local control rates [2,3,4,5,6,7,8,9,10].

The primary goals in early-stage glottic cancer treatment are favorable oncological outcomes and laryngeal function preservation, including voice, breathing, and swallowing. RT and TLM are standard treatments, with RT often preferred for better vocal outcomes [4,11]. TLM is ideal for superficial lesions, but deeper lesions may require adjuvant RT [3,11].due to the risk of incomplete resection. As a result, RT is commonly the first-line treatment, with surgery reserved for salvage therapy.

Evidence indicates that RT achieves over 90% 5-year locoregional control and survival rates for stage I disease, particularly with hypo-fractionated regimens. However, the outcomes for stage II disease are less favorable, even with altered fractionation [12,13,14,15,16,17,18,19,20,21]. This study aims to compare clinical outcomes of RT, prior microsurgery for tumor debulking, and concurrent chemoradiotherapy, while assessing tumor factors, such as vocal cord mobility, subglottic extension, and tumor differentiation, as prognostic indicators.

## 2. Materials and Methods

### 2.1. Patient Characteristics

From January 2001 to December 2022, 121 patients with T2 glottic carcinoma, treated with radiotherapy (RT), with or without cisplatin-based chemotherapy, were included in this study. Forty-four patients (36.4%) underwent transoral laser microsurgery (TLM) prior to RT or concurrent chemoradiotherapy (CCRT) for tumor debulking or tissue sampling. Patient data—including age, sex, performance status, smoking history, and tumor characteristics, such as pathological diagnosis, grade, stage, extent of tumor invasion, vocal cord mobility, and treatment modalities—were collected from medical records and summarized in Table 1.

The cohort consisted of 115 males and 6 females, with a median age of 62 years (range: 34–84). Regarding medical comorbidities, 25 patients had diabetes mellitus, 38 had hypertension, 4 had dyslipidemia, 6 had heart disease, and 4 had cerebrovascular disease. None had a history of cancer or synchronous malignancies. Most patients demonstrated good performance status (ECOG 0–1), except one with a score of 2. Additionally, 103 patients (85%) were current or former smokers. The follow-up duration ranged from 1 to 22 years, with a median of 8 years.

Pre-treatment evaluations included medical history, clinical assessment, laryngoscopy, biopsy, and imaging studies, such as Magnetic Resonance Imaging (MRI), Computed Tomography (CT), and Positron Emission Tomography (PET). Disease staging followed the AJCC system (7th edition). Of the 121 patients, 92 (76%) had T2a disease, characterized by the absence of impaired vocal cord mobility, and 29 (24%) had T2b disease, which included impaired vocal cord mobility as observed through endoscopic examination. Laryngoscopy/MRI/CT revealed that 65 patients (54%) had unilateral lesions, and 56 (46%) had bilateral vocal cord involvement. Anterior commissure involvement was noted in 99 patients (82%). Tumor extension included the supraglottic region in 70 patients (58%), subglottic extension in 24 (20%), and both regions in 24 (20%). None had nodal involvement or distant metastasis at diagnosis (Table 1).

### 2.2. Treatment

Total radiation doses administered ranged from 60 Gy to 76 Gy, with a median of 70 Gy. Fraction sizes varied between 1.8 Gy and 2.25 Gy, with a median of 2.12 Gy. To account for differences in radiobiological effects due to varying fractionation schemes, the Equivalent Dose in 2 Gy fractions (EQD2) was used for dose conversion. EQD2 was calculated using the formula: EQD2 = total dose × [dose per fraction + (α/β)]/[2 + (α/β)], with an α/β ratio of 10 Gy for laryngeal malignancies, in accordance with previous studies [22,23,24,25]. The most frequent fractionation regimens less than or equal to 70 Gy EQD2 were 66 Gy in 30 fractions for 16 patients, 70 Gy in 35 fractions for 10 patients and 66 Gy in 30 fractions for 9 patients. The most frequent fractionation regimens higher than 70 Gy EQD2 were 69.96 Gy in 33 fractions for 22 patients, 72 Gy in 36 fractions for 21 patients, and 72.6 Gy in 33 fractions for 11 patients. Only 4 patients with T2a disease received a scheduled radiation dose less than 6525 cGy.

In terms of irradiated fields and volumes, 50 patients received radiation only to the local tumor and entire larynx, while 71 patients received additional coverage of the cervical lymphatic regions. Specifically, 62 patients had bilateral coverage of levels II to IV, 7 had bilateral coverage limited to level III, and 2 had unilateral coverage of levels II to IV. Concurrent cisplatin-based chemotherapy was administered to 72 patients (59.5%), and 44 patients (36.4%) underwent prior transoral laser microsurgery (TLM). The overall treatment duration ranged from 38 to 108 days, with a median of 48 days, most completed within 70 days, though one patient had a prolonged duration of 108 days due to an episode of cerebrovascular infarction.

Acute treatment-related reactions, such as mucositis, laryngitis, pharyngitis, and dermatitis, were evaluated by radiation oncologists and otolaryngologists. After treatment, follow-up visits were scheduled every 3–4 months for the first two years, every 4–6 months between the third and fifth years, and annually thereafter. Follow-up evaluations included laryngoscopy, MRI or CT scans to monitor for tumor recurrence, complications, and the development of second malignancies.

### 2.3. Statistical Analysis

The endpoints for statistical analysis included locoregional (LR) control, recurrence-free survival (RFS), overall survival (OS), and failure patterns. Locoregional control was defined as the interval from treatment initiation to recurrence, patient death, or the last follow-up. RFS was defined as the time from diagnosis to recurrence or death, while OS was defined as the time from diagnosis to patient death or the last follow-up.

All endpoints were analyzed using the Kaplan–Meier method and actuarial survival analysis. Multivariate analysis was conducted using logistic regression to identify independent prognostic factors. Statistical significance was assessed with the log-rank test, and *p*-values less than 0.05 were considered statistically significant. All analyses were performed using SAS version 9.4 (SAS Inc., Cary, NC, USA).

## 3. Results

### 3.1. Locoregional Control and Survivals

The 5-year and 10-year LR control rates for the 121 patients were 82.1% and 77.8%, respectively. The RFS rates at 5 and 10 years were 72.9% and 58.3%, while OS rates were 80.2% and 66.8% at 5 and 10 years, respectively.

Among the 92 patients with T2a disease, the 5-year, 10-year LR controls, 10-year RFS, and 10-year OS rates were 85.5%, 79.8%, 58.7%, and 68.4%, respectively. In comparison, the 5-year, 10-year LR controls, 10-year RFS, and 10-year OS rates for the 29 patients with T2b disease were 70.8%, 70.8%, 56.2%, and 61.7%, respectively. There were no significant differences in LR control (*p* = 0.446), RFS (*p* = 0.841), or OS (*p* = 0.864) between the T2a and T2b groups.

Multivariate analysis identified concurrent chemotherapy with radiotherapy as the only significant prognostic factor for LR control (adjusted hazard ratio: 0.30, *p* = 0.011). The 5-year and 10-year LR control rates for patients receiving CCRT were 88.5% and 83.2%, respectively, compared to 72.8% and 69.6% for those receiving RT alone. Figure 1A illustrates the recurrence rates for patients with and without concurrent chemotherapy. Other factors, including age, smoking status, vocal cord mobility impairment, subglottic extension, pathology grade, total radiation dose, irradiation field, and prior microsurgery, were not significant for LR control. Multivariate analysis results for factors associated with LR recurrence are presented in Table 2.

Subgroup analyses were conducted to evaluate the effects of CCRT versus RT alone for patients with subglottic extension (Figure 2A–C) and impaired cord mobility (Figure 2D–F). For patients with subglottic extension, the 5-year LR control rates were 83% and 65% for those with and without concurrent chemotherapy, respectively (*p* = 0.414). For patients with impaired cord mobility, the 5-year LR control rates were 88% and 40% for those with and without concurrent chemotherapy, respectively (*p* = 0.003). The improved LR control associated with CCRT may be attributed to its effect on these two unfavorable subgroups of patients.

Further multivariate analysis of RFS and OS revealed additional benefits of concurrent chemotherapy. The 10-year RFS rates for patients with and without concurrent chemotherapy were 64.3% and 49.1%, respectively (*p* = 0.005). Figure 1B shows the RFS curve for patients with and without concurrent chemotherapy. Although total radiation dose was not statistically significant for RFS, patients receiving a total dose exceeding EQD2 70 Gy exhibited poorer OS (*p* = 0.012). The 10-year OS rates for patients with and without concurrent chemotherapy were 74.1% and 57.2%, respectively (*p* = 0.024). Figure 1C illustrates the OS curve for patients with and without concurrent chemotherapy.

Patient age over 60 years was an unfavorable factor for both RFS (*p* = 0.012) and OS (*p* = 0.002). Other factors, such as non-smoking status, absence of subglottic extension, normal vocal cord mobility, well-to-moderately differentiated pathology grade, irradiated field covering neck lymphatics, and prior TLM, were not significantly associated with improved RFS or OS. Multivariate analysis results for factors associated with RFS and OS are summarized in Table 3 and Table 4, respectively.

### 3.2. Failure Pattern

Of the 121 patients, 24 (19.8%) experienced locoregional recurrence, with the earliest occurring approximately 4 months post-radiotherapy. Among these, 20 patients presented with local lesions, 2 patients exhibited regional recurrence, and 2 patients experienced both local and regional recurrence. The median interval to recurrence was 17 months.

Salvage treatments, including partial or total laryngectomy, neck dissection, chemotherapy, and CCRT, were administered to most patients. However, 4 patients did not receive salvage treatment due to synchronous distant metastatic disease or poor health conditions.

Additionally, 9 patients (7.4%) developed distant metastasis. All cases involved the mediastinum or lungs (9/9), with 3 patients (33.3%) having bone metastasis and 1 patient (11.1%) exhibiting liver metastasis. At diagnosis, distant metastasis was isolated in 2 patients (22.2%), occurring without prior or concurrent locoregional recurrence. Three patients (33.3%) had a history of locoregional recurrence before distant metastasis, and 4 patients (44.4%) experienced locoregional recurrence concurrently with distant metastasis.

### 3.3. Adverse Events

All patients in this study experienced acute side effects of varying severity during radiotherapy (Table 5). Specifically, 10 patients (9.9%) developed Radiation Therapy Oncology Group (RTOG) grade III dermatitis, requiring wound care. Acute mucositis of grade II or higher was observed in a significant proportion of patients, with a markedly higher incidence among those receiving CCRT (*p* < 0.001). Additionally, 90 patients (74.4%) presented with RTOG grade II laryngitis, requiring steroid therapy for laryngeal edema, and 17 patients (14%) developed grade III laryngitis, necessitating hospitalization. Acute grade III pharyngitis occurred in 30 patients (24.8%), requiring opioid analgesics to manage odynophagia. The incidence of grade III acute pharyngitis and laryngitis was significantly higher in the CCRT group (*p* < 0.001 and *p* = 0.031, respectively).

Hematologic adverse events, including leukopenia, anemia, and thrombocytopenia, as well as transient renal or hepatic function impairments, were significantly more common in the CCRT group (*p* < 0.01). Importantly, no patients experienced grade IV or V acute toxicities during radiotherapy.

According to the RTOG late complication scale, 3 patients (2.5%) developed grade III laryngitis with vocal cord palsy, requiring tracheostomy, with one case requiring emergency intervention. However, there was no significant difference in late toxicities between patients receiving RT alone and those undergoing CCRT.

Records of adverse effects and causes of death revealed that all cases of grade III or higher late laryngitis occurred in patients who received a total radiation dose equivalent to EQD2 above 70 Gy. Causes of death included tumor recurrence, distant metastasis, pneumonia (potentially due to aspiration), and bleeding events, such as carotid artery rupture. These findings suggest that higher radiation doses may not necessarily improve outcomes and may instead contribute to long-term complications, including impaired swallowing and vascular damage within the irradiated region.

### 3.4. Second Malignancies

During the follow-up, 22 patients developed second malignancies, with a median diagnosis time of 74.5 months after diagnosis of laryngeal cancer. Thirteen cases (59.1%) were diagnosed over 5 years after the initial diagnosis of laryngeal malignancies. The second malignancies included 1 case of oral cavity cancer, 3 cases of oropharyngeal cancer, 2 cases of esophageal cancer, 7 cases of lung cancer, 2 cases of hepatocellular carcinoma, 4 cases of colorectal cancer, 1 case of prostate cancer, and 2 cases of acute myeloid leukemia.

## 4. Discussion

Early-stage glottic cancer (T1 and T2 stages) is typically treatable with RT alone, yielding high local control and survival rates. Recent advancements, including altered fractionation RT regimens, such as hypofractionation with increased fraction size and reduced overall treatment duration, have been shown to improves local control, particularly in T1 disease, but the outcomes remain less favorable for T2 disease, especially in T2b cases with hypomobile vocal cords [12,13,14,15,16,17]. Our previously published report revealed worse 10-year tumor control rates for T2 disease compared to T1 disease, with 10-year local control rates of 55% initially and 67% after salvage therapy, particularly in patients with subglottic extension, despite a median radiation dose of 70 Gy (range: 63–72.8 Gy) [13]. To improve outcomes, our department explored increasing radiation doses, with over 95% of patients receiving doses greater than the recommended 65.25 Gy for T2 glottic cancer. Another strategy was the addition of concurrent cisplatin chemotherapy.

Numerous studies have evaluated treatment outcomes for T2 stage glottic cancer using hyper-fractionated RT, hypo-fractionated RT, and CCRT(see Table 6). The RTOG 9512 trial compared hyper-fractionated RT (79.2 Gy in 66 fractions, twice daily) with conventional RT (70 Gy in 35 fractions, once daily). While hyper-fractionated RT showed trends toward improved 5-year local control (78% vs. 70%), disease-free survival (49% vs. 40%), and overall survival (72% vs. 63%), none were statistically significant [26] and resulted in higher rates of acute toxicities due to higher EQD2 (73.9 Gy).

Seno et al. reported a 10-year local control rate of 81% with hyper-fractionated RT (74.4 Gy in 62 fractions) [27]. Dixon et al. analyzed hypo-fractionated RT (52.5 Gy in 16 fractions) and reported 5-year local control, overall survival, and disease-specific survival rates of 82%, 67%, and 90%, respectively, with severe toxicity in only 1.8% of cases [23]. Motegi et al. achieved 5-year local control and overall survival rates of 77% and 91% using - RT (64.8 Gy in 27 fractions), with minimal severe toxicity [18]. De Ridder et al. found a 5-year local control rate of 77% with hypo-fractionated RT (60 Gy in 25 fractions) [21]. Overall, RT alone achieved 5-year local control rates of 77–81% for T2 glottic cancer, whether hyper-fractionated or hypo-fractionated.

In terms of CCRT, Akimoto et al.’s study of 50 T2 glottic cancer patients found significantly higher 5-year disease-free survival with CCRT (89%) compared to RT alone (68%) [28]. Kimura et al. reported 3-year local control, overall survival, and disease-specific survival rates of 96%, 95.5%, and 95.7%, respectively, in 31 T2 patients treated with concurrent chemotherapy and RT tailored to T2a (60 Gy in 30 fractions) and T2b (70 Gy in 35 fractions) [29]. Similarly, Ono et al. found 3-year local control rates of 93.2% for T2a and 85.7% for T2b using concurrent chemotherapy with RT doses of 60 Gy for T2a and 66–70 Gy for T2b [30]. Overall, CCRT improved local control (85.7–96%) compared to RT alone. Our study confirmed these findings, showing significant improvements in local-regional control (*p* = 0.011), recurrence-free survival (*p* = 0.005), and overall survival (*p* = 0.024) with cisplatin-based CCRT. The 3-year and 5-year local-regional control rates for CCRT were 91.6% and 88.5%, with 3-year and 5-year overall survival rates of 95.8% and 86.9%. Acute toxicities, including mucositis, pharyngitis, and hematologic effects, were more frequent in CCRT but manageable with supportive care. No significant differences in late complications were observed between RT and CCRT groups, though three cases of severe late laryngitis were linked to high radiation doses (EQD2 > 70 Gy). Thus, caution is advised when using high-dose RT in CCRT for T2 glottic cancer to minimize severe late toxicities. However, the Common Terminology Criteria for Adverse Events (CTCAE) was not employed to assess toxicity in this retrospective study, which primarily focused on radiotherapy-related toxicity. The observed toxicity rates may have been higher had CTCAE grading been utilized.

Tumors with subglottic extension (>5 mm below the true vocal cords) are linked to poorer outcomes [12,13,31,32]. Studies, including those by Le et al. [12] and Kim et al. [31], report significantly lower local control and 5-year disease-free survival in such cases. Our prior analysis also identified subglottic extension as a negative prognostic factor for local control (*p* = 0.02) [13], though this was not independently significant in our current study due to a small sample size. Patients with subglottic extension had lower 5-year locoregional control (76.5% vs. 85.8%), but outcomes improved with CCRT (from 65% to 83%).

T2b stage disease, characterized by impaired vocal cord mobility, also indicates worse prognosis due to tumor invasion into critical structures [16,17,23,33,34,35]. Studies, including Harwood et al. [33] and Le et al. [12], report significantly lower local control rates (e.g., 79% vs. 45% for normal vs. impaired mobility). Our study showed a marked improvement in 5-year locoregional control for T2b disease with concurrent chemotherapy (40% to 88%; *p* = 0.003), underscoring the benefit of CCRT.

Elective nodal irradiation is rarely used for early-stage glottic cancer due to low nodal metastasis risk (<2% for T2) [15]. Our study revealed local recurrence as the primary failure pattern, with only four cases of regional recurrence, supporting evidence that glottic tumors have distinct biological behavior compared to supra- or subglottic cancers [25]. Nodal coverage did not significantly impact locoregional control or survival outcomes.

This retrospective study on T2 glottic cancer highlights limitations, such as selection bias and heterogeneous radiotherapy regimens. To determine the optimal treatment strategy for T2 stage glottic cancer, further large-scale prospective clinical trials are warranted.

## 5. Conclusions

The study retrospectively analyzed T2 stage glottic cancers treated with various radiotherapy regimens, with or without concurrent cisplatin-based chemotherapy. The addition of concurrent cisplatin-based chemotherapy significantly enhanced locoregional control, progression-free survival, and overall survival, even yielding more favorable outcomes than prior studies. Acute toxicities were higher with concurrent chemoradiotherapy but did not increase late complications. Overall, concurrent cisplatin-based chemoradiotherapy was effective and safe for treating T2 glottic malignancies, though prospective trials or large-scale studies are needed to determine the optimal treatment approach for T2 glottic cancer.

## Figures and Tables

**Figure 1 cancers-17-00712-f001:**
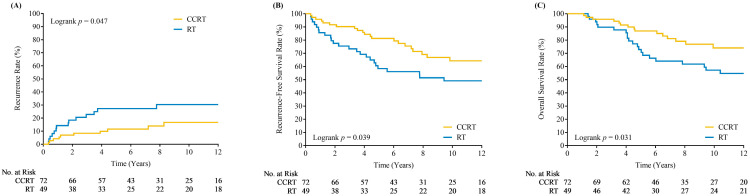
Kaplan–Meier plots of loco-regional recurrences (**A**), recurrence free survivals (**B**) and overall survivals (**C**) for patients who underwent concurrent chemoradiotherapy (CCRT) versus radiotherapy alone (RT).

**Figure 2 cancers-17-00712-f002:**
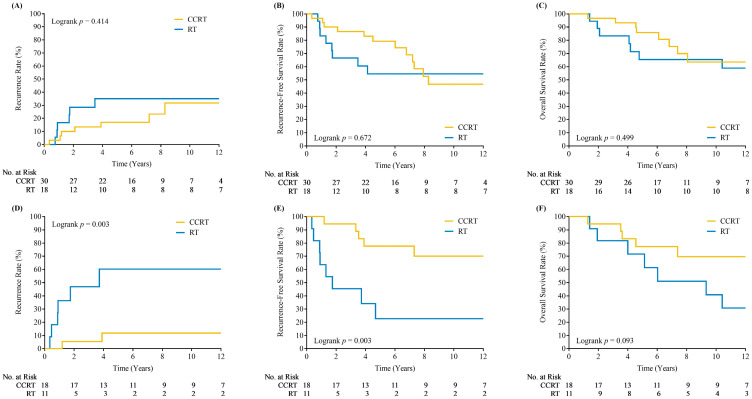
Kaplan–Meier plots of loco-regional recurrences, recurrence free survivals and overall survivals for patients with subglottic extension (**A**–**C**) and impaired cord mobility (**D**–**F**) who underwent concurrent chemoradiotherapy (CCRT) versus radiotherapy alone (RT).

**Table 1 cancers-17-00712-t001:** Characteristics of the study population.

Variables	No	%
Total	121	
Age		
<60	43	35.5
≥60	78	64.5
Sex		
Male	115	95.0
Female	6	5.0
Smoking		
No	18	14.9
Yes	103	85.1
Supraglottic or Subglottic extension		
Supraglottic	70	57.9
Subglottic	24	19.8
Both	24	19.8
No	3	2.5
Impaired vocal fold mobility		
No	92	76.0
Yes	29	24.0
Laterality		
Unilateral	65	53.7
Bilateral	56	46.3
Anterior commissure involvement		
No	22	18.2
Yes	99	81.8
Grade		
Well to moderately differentiated	107	88.4
Poorly differentiated	14	11.6
EQD_2_, cGy		
<7000	45	37.2
≥7000	76	62.8
RT field		
Larynx only	50	41.3
Larynx + elective nodal RT	71	58.7
TLM		
No	77	63.6
Yes	44	36.4
CCRT		
No	49	40.5
Yes	72	59.5

EQD_2_: Equivalent Dose in 2 Gy fractions (EQD_2_); RT: Radiotherapy; TLM: Transoral laser microsurgery; CCRT: Concurrent chemo-radiotherapy.

**Table 2 cancers-17-00712-t002:** Multivariable Cox proportional hazards model analysis of factors associated with recurrence.

	Crude HR		Adjusted HR	
Variables	HR (95% CI)	*p*-Value	HR (95% CI)	*p*-Value
Age				
<60	Reference		Reference	
≥60	1.53 (0.63–3.69)	0.347	1.32 (0.52–3.33)	0.562
Smoking				
No	Reference		Reference	
Yes	2.07 (0.49–8.81)	0.325	2.28 (0.49–10.5)	0.291
Subglottic				
No	Reference		Reference	
Yes	1.92 (0.86–4.29)	0.112	1.57 (0.66–3.77)	0.309
Impaired vocal fold morbidity				
No	Reference		Reference	
Yes	1.77 (0.76–4.14)	0.188	1.37 (0.54–3.47)	0.514
Grade				
Well to moderately differentiated	Reference		Reference	
Poorly differentiated	0.68 (0.16–2.91)	0.607	0.89 (0.19–4.17)	0.877
EQD_2_, cGy				
<7000	Reference		Reference	
≥7000	1.57 (0.65–3.80)	0.315	2.15 (0.77–6.03)	0.146
RT field				
Larynx only	Reference		Reference	
Larynx + elective nodal RT	0.98 (0.43–2.20)	0.956	1.21 (0.50–2.91)	0.674
TLM				
No	Reference		Reference	
Yes	0.53 (0.21–1.35)	0.183	0.71 (0.26–1.94)	0.501
CCRT				
No	Reference		Reference	
Yes	0.45 (0.20–1.01)	0.053	0.30 (0.12–0.76)	0.011

HR: Hazard ratio; CI: Confidence interval; EQD_2_: Equivalent Dose in 2 Gy fractions (EQD_2_); RT: Radiotherapy; TLM: Transoral laser microsurgery; CCRT: Concurrent chemo-radiotherapy.

**Table 3 cancers-17-00712-t003:** Multivariable Cox proportional hazards model analysis of factors associated with recurrence-free survival.

	Crude HR		Adjusted HR	
Variables	HR (95% CI)	*p*-Value	HR (95% CI)	*p*-Value
Age				
<60	Reference		Reference	
≥60	2.79 (1.34–5.81)	0.006	2.46 (1.15–6.27)	0.021
Smoking				
No	Reference		Reference	
Yes	1.22 (0.52–2.89)	0.644	1.18 (0.48–2.91)	0.720
Subglottic				
No	Reference		Reference	
Yes	1.37 (0.76–2.47)	0.293	1.22 (0.65–2.30)	0.537
Impaired vocal fold morbidity				
No	Reference		Reference	
Yes	1.51 (0.79–2.89)	0.208	1.44 (0.71–2.88)	0.311
Grade				
Well to moderately differentiated	Reference		Reference	
Poorly differentiated	0.97 (0.38–2.45)	0.942	1.16 (0.42–3.19)	0.771
EQD_2_, cGy				
<7000	Reference		Reference	
≥7000	1.26 (0.68–2.33)	0.456	1.77 (0.85–3.68)	0.124
RT field				
Larynx only	Reference		Reference	
Larynx + elective nodal RT	0.88 (0.49–1.58)	0.665	1.07 (0.57–2.02)	0.835
TLM				
No	Reference		Reference	
Yes	0.51 (0.26–1.00)	0.050	0.62 (0.30–1.30)	0.204
CCRT				
No	Reference		Reference	
Yes	0.54 (0.30–0.98)	0.042	0.45 (0.23–0.89)	0.021

HR: Hazard ratio; CI: Confidence interval; EQD_2_: Equivalent Dose in 2 Gy fractions (EQD_2_); RT: Radiotherapy; TLM: Transoral laser microsurgery; CCRT: Concurrent chemo-radiotherapy.

**Table 4 cancers-17-00712-t004:** Multivariable Cox proportional hazards model analysis of factors associated with overall survival.

	Crude HR		Adjusted HR	
Variables	HR (95% CI)	*p*-Value	HR (95% CI)	*p*-Value
Age				
<60	Reference		Reference	
≥60	5.95 (2.10–16.9)	0.001	5.24 (1.78–15.5)	0.003
Smoking				
No	Reference		Reference	
Yes	0.85 (0.35–2.05)	0.720	0.69 (0.27–1.78)	0.447
Subglottic				
No	Reference		Reference	
Yes	1.20 (0.62–2.32)	0.598	1.06 (0.53–2.13)	0.869
Impaired vocal fold morbidity				
No	Reference		Reference	
Yes	1.71 (0.85–3.42)	0.130	1.20 (0.56–2.55)	0.646
Grade				
Well to moderately differentiated	Reference		Reference	
Poorly differentiated	1.05 (0.37–2.98)	0.925	1.41 (0.46–4.36)	0.548
EQD_2_, cGy				
<7000	Reference		Reference	
≥7000	1.69 (0.83–3.44)	0.149	2.73 (1.17–6.36)	0.020
RT field				
Larynx only	Reference		Reference	
Larynx + elective nodal RT	0.85 (0.44–1.65)	0.634	0.93 (0.46–1.88)	0.843
TLM				
No	Reference		Reference	
Yes	0.47 (0.21–1.03)	0.059	0.67 (0.28–1.60)	0.362
CCRT				
No	Reference		Reference	
Yes	0.49 (0.25–0.95)	0.034	0.45 (0.21–0.95)	0.035

HR: Hazard ratio; CI: Confidence interval; EQD_2_: Equivalent Dose in 2 Gy fractions (EQD_2_); RT: Radiotherapy; TLM: Transoral laser microsurgery; CCRT: Concurrent chemo-radiotherapy.

**Table 5 cancers-17-00712-t005:** Acute and Late Toxic effects, according to the treatment groups.

Variables	RT		CCRT		*p*-Value
	N = 49	(%)	N = 72	(%)	
Acute toxicities					
Acute dermatitis					0.116
0	3	(6.1)	3	(4.2)	
1	28	(57.1)	30	(41.7)	
2	17	(34.7)	30	(41.7)	
3	1	(2.0)	9	(12.5)	
Acute mucositis					<0.001
0	30	(61.2)	8	(11.1)	
1	13	(26.5)	17	(23.6)	
2	5	(10.2)	38	(52.8)	
3	1	(2.0)	9	(12.5)	
Acute pharyngitis					<0.001
1	17	(34.7)	9	(12.5)	
2	28	(57.1)	37	(51.4)	
3	4	(8.2)	26	(36.1)	
Acute laryngitis					0.031
1	7	(14.3)	7	(9.7)	
2	40	(81.6)	50	(69.4)	
3	2	(4.1)	15	(20.8)	
Nause/Vomiting					0.025
0	47	(95.9)	55	(76.4)	
1	2	(4.1)	7	(9.7)	
2	0	(0)	9	(12.5)	
3	0	(0)	1	(1.4)	
Leukopenia					<0.001
0	49	(100)	32	(44.4)	
1	0	(0)	22	(30.6)	
2	0	(0)	13	(18.1)	
3	0	(0)	5	(6.9)	
Anemia					<0.001
0	49	(100)	17	(23.6)	
1	0	(0)	41	(56.9)	
2	0	(0)	12	(16.7)	
3	0	(0)	2	(2.8)	
Thrombocytopenia					<0.001
0	49	(100)	28	(38.9)	
1	0	(0)	38	(52.8)	
2	0	(0)	4	(5.6)	
3	0	(0)	2	(2.8)	
Renal function					<0.001
0	49	(100)	50	(69.4)	
1	0	(0)	19	(26.2)	
2	0	(0)	1	(1.4)	
3	0	(0)	2	(2.8)	
Liver function					0.005
0	49	(100)	58	(80.6)	
1	0	(0)	11	(15.3)	
2	0	(0)	3	(4.2)	
Late toxicities					
Vocal cord palsy					0.954
0	40	(81.6)	55	(76.4)	
1	4	(8.2)	9	(12.5)	
2	2	(4.1)	3	(4.2)	
3	2	(4.1)	3	(4.2)	
4	1	(0.0)	2	(2.8)	
Laryngeal edema					0.482
0	5	(10.2)	4	(5.6)	
1	37	(75.5)	50	(69.4)	
2	6	(12.2)	12	(16.7)	
3	1	(2.0)	5	(6.9)	
4	0	(0)	1	(1.4)	
Laryngeal necrosis					0.179
0	44	(89.8)	67	(93.1)	
1	3	(6.1)	1	(1.4)	
2	2	(4.1)	1	(1.4)	
3	0	(0)	3	(4.2)	
* Chronic aspiration					0.065
0	43	(87.8)	68	(94.4)	
1	6	(12.2)	2	(2.8)	
2	0	(0)	2	(2.8)	
Neck fibrosis					0.367
0	43	(87.8)	56	(77.8)	
1	5	(10.2)	14	(19.4)	
2	1	(2.0)	2	(2.8)	
Pneumonia					0.514
yes	0	(0)	2	(2.9)	
no	49	(100)	70	(97.1)	
Tracheostomy					0.271
yes	0	(0)	3	(4.2)	
no	49	(100)	69	(95.8)	
Laryngectomy					0.514
yes	0	(0)	2	(2.9)	
no	49	(100)	70	(97.1)	
Gastric tube dependency > 3 months					0.407
yes	0	(0)	1	(1.4)	
no	49	(100)	71	(98.6)	
Death due to laryngeal dysfunction					NA
yes	0	(0)	0	(0)	
no	49	(100)	72	(100)	

RT: radiotherapy alone, CCRT: concurrent chemo-radiotherapy. NA: not available. *Grade 1 chronic aspiration was defined as recurrent episodes of choking documented in two or more consecutive follow-up assessments after three months of treatment and without fever, dyspnea, productive cough or other significant infection signs. Grade 2 chronic aspiration was defined as recurrent episodes of choking documented in two or more consecutive follow-up assessments after three months of treatment, accompanied by radiographic evidence of infiltration on chest X-ray or CT scan, which require antibiotic treatment for aspiration pneumonia or necessitate nasogastric tube insertion to prevent further aspiration events. The grading system were modified from Common Terminology Criteria for Adverse Events (CTCAE) Version 5, grades of Gastrointestinal disorders—Other.

**Table 6 cancers-17-00712-t006:** The outcomes of various studies on T2 glottic cancer investigating the efficacy of different radiotherapy regimens, as well as the addition of concurrent chemotherapy.

Study	Patients	Treatment Modality	Outcome
Hyperfractionation Radiotherapy			
RTOG 9512 [26]	120 T2 patients	79.2 Gy/66 fractions, BID	5-year local control = 78% 5-year disease-free survival = 49% 5-year overall survival = 72%
Seno et al. [27]	66 T2 patients	74.4 Gy/62 fractions, BID	10-year local control = 81%
Hypofractionation Radiotherapy			
Dixon et al. [23]	112 T2 patients	52.5 Gy/16 fractions	5-year local control = 82% 5-year overall survival = 67%
Motegi et al. [18]	44 T2 patients	64.8 Gy/27 fractions	5-year local control = 77% 5-year overall survival = 91%
de Ridder et al. [21]	30 T2 patients	60 Gy/25 fractions	5-year local control = 77%
CCRT			
Akimoto et al. [28]	27 T2 patients	CCRT with Cisplatin/Docetaxel	5-year locoregional control = 89% 5-year disease-free survival = 89% 5-year overall survival = 96%
Kimura et al. [29]	31 T2 patients	CCRT with S-1 or Cisplatin	3-year local control = 96% 3-year overall survival = 95.5%
Ono et al. [30]	40 T2 patients	CCRT with S-1 or Cisplatin	3-year local control = 92%
Our study	72 T2 patients	CCRT with Cisplatin	3-year locoregional control = 91.6% 5-year locoregional control = 88.5% 10-year locoregional control = 83.2% 3-year overall survival = 95.8% 5-year overall survival = 86.9% 10-year overall survival = 74.1%

CCRT: Concurrent chemo-radiotherapy.

## Data Availability

The data that support the findings of this study are available on request from the corresponding authors. Access will be provided upon reasonable request and subsequent approval from the IRB of Chang Gung Memorial Hospital, Chiayi branch. The data are not publicly available due to privacy or ethical restrictions.

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
