# Peer review of "Outcome of T2 Glottic Cancer Treated with Radiotherapy Alone or Concurrent Chemo-Radiotherapy"

_cancers, 2025, doi:10.3390/cancers17040712_

Round 1
Reviewer 1 Report
Comments and Suggestions for Authors
The authors in this study investigated a very important clinical question. The outcome for T2 glottic cancer with radiotherapy is very suboptimal. Improvement in outcome is needed.
1) As this article is about chemotherapy, please put supplementary table to the main manuscript and put table 3 and/or table 5 to supplementary section.
2) What is the definition of "chronic aspiration" in the supplementary table.
3) Please add deaths, pneumonia, tracheostomy, g-tube dependency (>3 months after treatment), and laryngectomy due to dysfunctional larynx in the supplementary table.
4) In the abstract, please replace "higher acute toxicity" with "high toxicity" in the conclusion.
5) The authors should mention in the discussion section that CTCAE was not used to grade toxicity, the actual toxicity rates could be higher. CTCAE is more comprehensive than RTOG toxicity grading and has been the standard grading system for clinical trials.
Author Response
Thanks for the comments from reviewer 1.
1) As this article is about chemotherapy, please put supplementary table to the main manuscript and put table 3 and/or table 5 to supplementary section.
Ans: Thanks for the kindly advice from the reviewer. Since there is no limitation in the numbers of Tables in the Cancers journal. I will add the supplementary table to Table 5 and change the Table 5 into Table 6.
2) What is the definition of "chronic aspiration" in the supplementary table.
Ans: The definitions of chronic aspiration were based on the Common Terminology Criteria for Adverse Events (CTCAE) Version 5, grades of Gastrointestinal disorders – Other. The definitions of grade 1 & grade 2 toxicities were added into the footnotes of new Table 5.
“* Grade 1 chronic aspiration was defined as recurrent episodes of choking documented in two or more consecutive follow-up assessments after three months of treatment and without fever, dyspnea, productive cough or other significant infections sign. Grade 2 chronic aspiration as recurrent episodes of choking documented in two or more consecutive follow-up assessments after three months of treatment, accompanied by radiographic evidence of infiltration on chest X-ray or CT scan, which require antibiotic treatment for aspiration pneumonia or necessitate nasogastric tube insertion to prevent further aspiration events. The grading system were modified from Common Terminology Criteria for Adverse Events (CTCAE) Version 5, grades of Gastrointestinal disorders – Other.”
3) Please add deaths, pneumonia, tracheostomy, g-tube dependency (>3 months after treatment), and laryngectomy due to dysfunctional larynx in the supplementary table.
Ans: Thank you for pointing this out. I agree with this comment. Therefore, I have added the above-mentioned items into new Table 5.
Pneumonia |
|
|
|
|
|
0.514 |
yes |
|
0 |
(0) |
2 |
(2.9) |
|
no |
|
49 |
(100) |
70 |
(97.1) |
|
Tracheostomy |
|
|
|
|
|
0.271 |
yes |
|
0 |
(0) |
3 |
(4.2) |
|
no |
|
49 |
(100) |
69 |
(95.8) |
|
Laryngectomy |
|
|
|
|
|
0.514 |
yes |
|
0 |
(0) |
2 |
(2.9) |
|
no |
|
49 |
(100) |
70 |
(97.1) |
|
Gastric tube dependency > 3 months |
|
|
|
|
|
0.407 |
yes |
|
0 |
(0) |
1 |
(1.4) |
|
no |
|
49 |
(100) |
71 |
(98.6) |
|
Death due to laryngeal dysfunction |
|
|
|
|
|
NA |
yes |
|
0 |
(0) |
0 |
(0) |
|
4) In the abstract, please replace "higher acute toxicity" with "high toxicity" in the conclusion.
Ans: Agree. I have replaced "higher acute toxicity" with "high toxicity" in the conclusion section.
Line 46-47
“Conclusions: CCRT with cisplatin improves local control, recurrence-free survival, and overall survival in T2N0M0 glottic cancer, albeit with high toxicity.”
5) The authors should mention in the discussion section that CTCAE was not used to grade toxicity, the actual toxicity rates could be higher. CTCAE is more comprehensive than RTOG toxicity grading and has been the standard grading system for clinical trials.
Ans: I agree with this and have incorporated your suggestion into the discussion section.
Line 311-314
“However, the Common Terminology Criteria for Adverse Events (CTCAE) was not employed to assess toxicity in this retrospective study, which primarily focused on radiotherapy-related toxicity. The observed toxicity rates may have been higher had CTCAE grading been utilized.”
Reviewer 2 Report
Comments and Suggestions for Authors
This is a very interesting study addressing an important point about the value of adding concurrent chemotherapy to radiotherapy for T2 larynx cancer. The study appears well conducted and well presented, and contrasts with the standard recommendation of just using chemoradiotherapy for T2 glottic cancers with adverse features, namely subglottic extension or impaired cord mobility.
There are a few points that might be addressed to help your overall conclusions.
You have looked at biologically equivalent doses expressed as EQD2, which is fine in addressing fractionation, but does not address overall time, which in head and neck cancer is well recognised as a determinant of outcome. You do mention one case where the overall time was 108 days, rendering that particular treatment potentially less effective. I would recommend reanalysing your data with BED calculated with an overall time correction
Your subgroup analysis is helpful in looking at the potential impact of chemoradiotherapy on outcome with subglottic extension and impaired cord mobility (lines 177-183; Figure 2), as the multivariate analysis had not identified these as adverse risk factors (Table 3), perhaps due to low patient numbers resulting in wide confidence intervals for the hazard ratios. Your readers may see this as the multivariate analysis as not supporting the traditional view, whereas the subgroup analysis does (although the possibility of confounding variables in simple subgroup analysis as compared to multivariate analysis cannot be ignored). Your argument would be strengthened by presenting the results with and without chemotherapy for those patients who did not have either risk factor (as an addition to Figure 2). If that showed no benefit from chemotherapy, that would support the traditional view. However, if a benefit persisted, and too the same degree as in the other subgroups, that would support more widespread use of chemotherapy in these patients.
References: a number of these where you cite the authors in the text are incorrect. For example line 287: "Lynne M" in the text is correctly cited as Dixon LM (reference 23), so should be quoted as "Dixon" in the text. Most of the references in the Discussion need correcting. In general, it is not necessary to mention the lead author's name in the text. Not doing that would be simpler.
Line 130: do you mean "radiation technologists and oncologists"?
Author Response
To Reviewer #2:
1) You have looked at biologically equivalent doses expressed as EQD2, which is fine in addressing fractionation, but does not address overall time, which in head and neck cancer is well recognised as a determinant of outcome. You do mention one case where the overall time was 108 days, rendering that particular treatment potentially less effective. I would recommend reanalysing your data with BED calculated with an overall time correction
Ans: Thank you for this valuable suggestion. It would have been interesting to explore this aspect. However, in the context of our study, the patient who received radiotherapy over a total treatment period of 108 days did not experience tumor recurrence. Among the 121 patients, only 7 had treatment delays exceeding 10 days. We have chosen not to re-analyze the data with time adjustments in this study, as it is unlikely to alter the primary findings.
2) Your subgroup analysis is helpful in looking at the potential impact of chemoradiotherapy on outcome with subglottic extension and impaired cord mobility (lines 177-183; Figure 2), as the multivariate analysis had not identified these as adverse risk factors (Table 3), perhaps due to low patient numbers resulting in wide confidence intervals for the hazard ratios. Your readers may see this as the multivariate analysis as not supporting the traditional view, whereas the subgroup analysis does (although the possibility of confounding variables in simple subgroup analysis as compared to multivariate analysis cannot be ignored). Your argument would be strengthened by presenting the results with and without chemotherapy for those patients who did not have either risk factor (as an addition to Figure 2). If that showed no benefit from chemotherapy, that would support the traditional view. However, if a benefit persisted, and too the same degree as in the other subgroups, that would support more widespread use of chemotherapy in these patients.
Ans: Thank you for highlighting this point. I agree with your comment. As a result, I have conducted additional analyses for patients without either subglottic extension or impaired cord mobility as risk factors (n = 60). The 5-year loco-regional recurrence rates (A), recurrence-free survival rates (B), and overall survival rates (C) were 9.7% vs. 15.1% (p = 0.308), 84.3% vs. 69.8% (p = 0.119), and 90.7% vs. 77.4% (p = 0.107) for patients with and without concurrent chemotherapy, respectively. Although there is a trend toward better outcomes for patients treated with concurrent chemoradiotherapy (CCRT), a larger sample size is necessary to draw definitive conclusions regarding this subgroup.
3) References: a number of these where you cite the authors in the text are incorrect. For example line 287: "Lynne M" in the text is correctly cited as Dixon LM (reference 23), so should be quoted as "Dixon" in the text. Most of the references in the Discussion need correcting. In general, it is not necessary to mention the lead author's name in the text. Not doing that would be simpler.
Ans: Thank you for your reminder. I had mistakenly labeled the references with the first names instead of the last names. I have reviewed all references and corrected them to include the appropriate citations.
4) Line 130: do you mean "radiation technologists and oncologists"?
Ans: I mean “by radiation oncologists and otolaryngologists.”
Line 130: “…by radiation oncologists and otolaryngologists.”